# PROMPT LEARNING WITH QUATERNION NETWORKS

**Boya Shi[1,2], Zhengqin Xu[1], Shuai Jia[1], Chao Ma[1]***
[1] MoE Key Lab of Artificial Intelligence, AI Institute, Shanghai Jiao Tong University
[2] National Innovation Institute of Defense Technology
{boya.shi, fate311, jiashuai, chaoma}@sjtu.edu.cn

## ABSTRACT

Multimodal pre-trained models have shown impressive potential in enhancing performance on downstream tasks. However, existing multimodality fusion strategies primarily rely on explicit interaction structures that fail to capture the diverse aspects and patterns inherent in input data. This yields limited performance in zero-shot contexts, especially when fine-grained classifications and abstract interpretations are required. To address this issue, we propose an effective approach, namely Prompt Learning with Quaternion Networks (QNet), for semantic alignment across diverse modalities. QNet employs a quaternion hidden space where the mutually orthogonal imaginary axes capture rich intermodal semantic spatial correlations from various perspectives. Hierarchical features across multilayers are utilized to encode intricate interdependencies within various modalities with a reduced number of parameters. Our experiments on 11 datasets demonstrate that QNet outperforms state-of-the-art prompt learning techniques in base-to-novel generalization, cross-dataset transfer, and domain transfer scenarios with fewer learnable parameters. The source code is available at https://github.com/SHIBOYA/QNet.

## 1 INTRODUCTION

Large-scale multimodal pre-trained models, such as ChatGPT (Radford et al., 2019) and its successor ChatGPT-4 (OpenAI, 2023), have recently shown great promise in the pursuit of general artificial intelligence. These models can facilitate various downstream tasks using prompts, which tap into the intrinsic knowledge embedded within them (Jin et al., 2021; Yao et al., 2021; Ding et al., 2022; Lüddecke & Ecker, 2022). However, it is still challenging for multimodal pre-trained models like CLIP (Radford et al., 2021) to handle more abstract tasks such as sketch recognition. This implies that existing pre-trained models rely heavily on homogeneous representation present in the training data for image classification, and are not effective in capturing the diverse and complementary features across different modalities. Therefore, it is of great importance to improve the modality fusion capabilities of pre-trained models within zero-shot settings, which is a significant and challenging task (Lu et al., 2022; Huang et al., 2022; Manli et al., 2022).

CoOp (Zhou et al., 2022b) is a technique that incorporates continuous prompt variables to improve the performance of CLIP in observable categories. Co-CoOp (Zhou et al., 2022a) introduces conditional prompt learning to enhance the generalization capability of CLIP in non-observable categories, enabling it to better acquire image features. While MaPLe (Khattak et al., 2022) is another method that enhances the collaboration between images and text through mapping, its linear connections assume a linear relationship among different modalities, which may not always hold true, potentially leading to interference between varying semantic spaces. Cross-attention mechanisms, which use non-linear fusion layers to merge features from diverse semantic spaces, may not effectively integrate complementary knowledge across modalities. As a result, both linear connections and other auxiliary networks struggle with homogeneous representations, making it difficult to capture distinct aspects or patterns within the data. To overcome this challenge, a fusion strategy should be formulated that can encapsulate the inherent intricacies in the relationships among diverse modalities.

Quaternion (Parcollet et al., 2018) is a method for representing high-dimensional data that offers unique perspectives on input features through its mutually orthogonal imaginary axes. By utilizing

---

*C. Ma is the corresponding author.

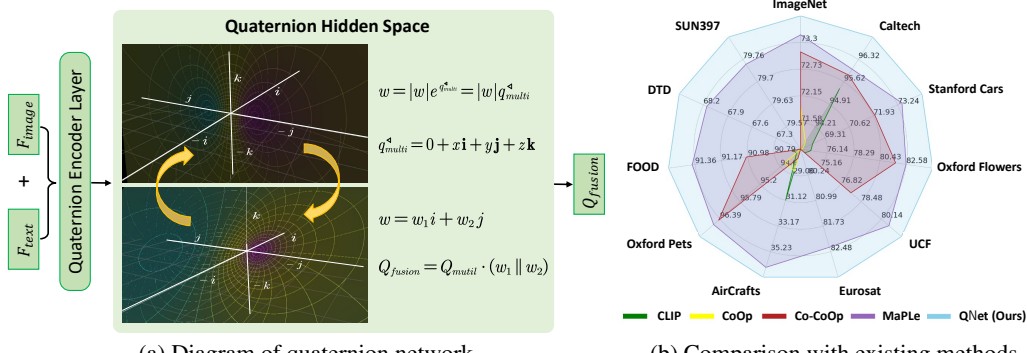

(a) Diagram of quaternion network      (b) Comparison with existing methods

Figure 1: (a) The quaternion encoder layer effectively incorporates multimodal features into the quaternion hidden space, utilizing three orthogonal imaginary axes to disentangle complex interrelationships among modalities. The k-axis plays a crucial role in balancing modalities' contributions, resulting in optimal performance. (b) Our QNet has achieved exceptional results in the novel class generalization task, outperforming other models on 11 datasets based on the harmonic mean (HM).

the inherent orthogonal relationship among the imaginary axes within the quaternion hidden space, correlations among diverse modal features can be established, which enables a more comprehensive representation within the shared semantic space. To this end, we propose Prompt Learning with Quaternion Networks (QNet) to better fuse the features among diverse modalities. In the quaternion hidden space, the three mutually orthogonal imaginary axes, namely $i$, $j$, and $k$, allocate unique weights to various distribution features from diverse perspectives, as demonstrated in Figure 1(a). By leveraging this fused feature representation, we can directly construct prompts for both visual and textual domains, which effectively circumvents the necessity of crafting explicit interaction architectures for modality fusion. In zero-shot scenarios, the strategic incorporation of QNet at the input stage of CLIP significantly outperforms Co-CoOp, yielding an absolute enhancement exceeding 10% on novel class datasets such as EuroSAT (Helber et al., 2019) and FGVC-Aircraft (Maji et al., 2013). Furthermore, our proposed method facilitates a deeper fusion of features originating from hierarchical levels with fewer trainable parameters (i.e., 2.93M) than MaPLe (i.e., 3.56M), thereby surpassing existing multimodal models across 11 datasets, as shown in Figure 1(b). In summary, we make the following contributions:

- We develop a novel approach to model complex relationships between different modalities in multimodal pre-trained models, even in situations where no data is available for training. Our method incorporates quaternion networks to fuse modalities more effectively than traditional methods. We first use quaternion networks in multimodal prompt learning.

- We use the mutually orthogonal multiple imaginary axes of quaternions to effectively disentangle complex feature distributions of different modalities in the latent feature space. Our proposed QNet is versatile and can be used for other multimodal fusion tasks, as it encapsulates interdependent relationships within features using compact parameters.

- To enhance the feature fusion potential of QNet, we utilize quaternion networks to combine features from different hierarchical levels. Extensive experiments on several datasets show that our approach consistently outperforms other multimodal models by a significant margin.

## 2 RELATED WORK

**Vision language models.** Multimodal fusion approaches heavily rely on the availability of large-scale multimodal datasets (Li et al., 2020; 2021; Jia et al., 2021; Zhai et al., 2022). For instance, CLIP (Radford et al., 2021) makes use of 400 million image-text pairs in its training process to achieve remarkable performance. However, utilizing these pre-trained models for downstream tasks with limited data (Gao et al., 2021; Zhang et al., 2022; Kim et al., 2022; Rasheed et al., 2022) remains a significant challenge in the field of multimodal data analysis, especially in terms of exploring the correlations between images and text (Maaz et al., 2022; Gu et al., 2021; Zang et al., 2022; Feng et al., 2022). Although ALBEF (Li et al., 2021) has made substantial progress in integrating image and text modalities through a six-layer transformer architecture, the issue of homogeneous representation

persists, making it difficult to capture distinct aspects or patterns within the data. In this work, we propose a novel image and text fusion architecture based on quaternion networks (Parcollet et al., 2018), which unravels the complex relationships between different modal distributions, with the aim of providing a more efficient approach to multimodal feature fusion.

**Prompt learning.** Advances in natural language processing (NLP) (Li & Liang, 2021; Lester et al., 2021; Liu et al., 2021) have expanded the use of prompts to computer vision tasks (Jia et al., 2022; Wang et al., 2022b;a). This has brought about new techniques such as CoOp (Zhou et al., 2022b), which utilizes continuous vectors to enhance performance on previously unseen classes by implementing instance-level prompts. Another technique, MaPLe (Khattak et al., 2022), intensifies modality interactions by mapping text prompts to image prompts, but struggles to create diverse prompts and convey semantic information through different modal features. Our proposed method leverages quaternions with mutually orthogonal imaginary axes to calculate the weights of multimodal fusion features from various perspectives, effectively separating the complex distributions among different modal features and facilitating the integration of complementary information within the latent space. This novel modality fusion strategy represents a significant contribution to existing research on multimodal learning.

**Zero-shot learning.** Zero-shot learning is a powerful technique that enables pre-trained models to recognize new categories. To achieve this, auxiliary information such as attributes or word embeddings (Chao et al., 2016; Wang et al., 2019b; Xian et al., 2017; Yi et al., 2022) is provided to the models. Research indicates that prompts can activate multimodal knowledge within pre-trained models, boosting their performance in zero-shot scenarios. For example, DALLE (Reddy et al., 2021) is a model that generates multimodal images using prompts and a conditional decoder, showing impressive results in zero-shot generation. Another model, CaFo (Zhang et al., 2023), uses multiple image-text contrastive loss functions to align image and text features through a collaborative training process. However, our research delves into a more fundamental question: can we acquire features that contain even richer multimodal knowledge to facilitate diverse downstream tasks for pre-trained multimodal models? Inspired by the quaternion network (Parcollet et al., 2018), we develop more efficient data structures that facilitate prompt learning and the acquisition of fruitful multimodal fusion features. This enhancement significantly improves the pre-trained models' capability to process and predict accurately across diverse and large-scale data distributions.

## 3 METHODOLOGY

### 3.1 MOTIVATION

Deep learning models face significant challenges in representing images as arrays of pixels, which is due to the complex task of accurately encoding dependencies among internal features within high-dimensional data spaces. The challenges are further compounded in multimodal scenarios where multiple modalities, such as text, audio, and images, need to be aligned and jointly processed. The intricate distributions of these modalities can interfere with each other, making it difficult to extract essential features. Prevailing methods for image-text scenarios often rely on sophisticated structures to align visual and textual content, which results in additional network parameters. These approaches contradict the objective of enhancing the performance of multimodal pre-trained models within zero-shot settings.

On the other hand, the elegant algebraic representation of quaternion offers a natural solution to these challenges. It can encode the dependencies between features and internal characteristics while reducing parameters. The three imaginary axes in the quaternion space provide three distinct perspectives for decoupling the intricate distributions of various modalities, effectively resolving the issue of homogeneous representations that fail to capture disparate aspects or patterns within the data.

Our QNet is inspired by the concept of quaternion, by which we utilize fewer parameters to capture the interdependence of multimodal features, achieving prompt learning efficiency. Our method seamlessly integrates with multimodal models and outperforms the existing MaPLe (Khattak et al., 2022) method on multiple datasets with merely one prompt layer. We further observe that adding more prompt layers improves the performance of quaternion networks. Additionally, we have incorporated quaternion networks into various layers of pre-trained models to leverage their potential in feature fusion. This incorporation allows for a more profound fusion of features across multiple modalities and hierarchical levels, while using fewer learnable parameters.

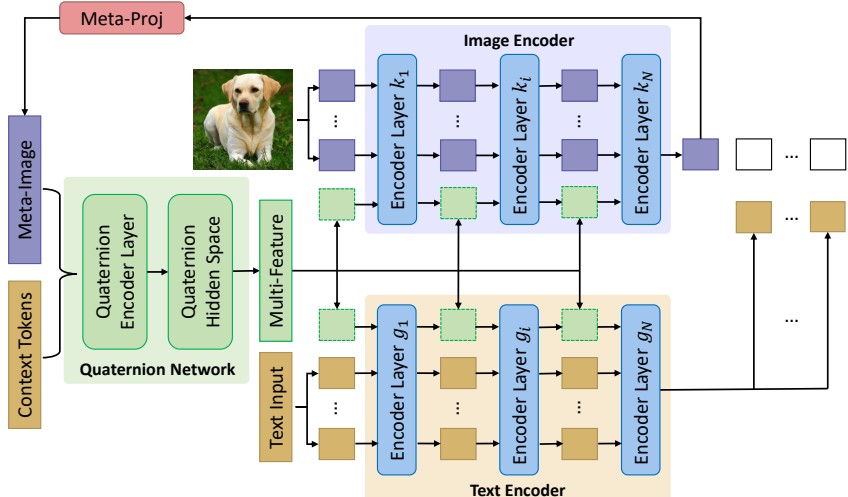

Figure 2: Overview of prompt learning with quaternion networks (QNet). The proposed QNet framework involves a sophisticated process that effectively transforms image features in purple through Meta-Proj. The transformed features are then combined with textual context tokens in yellow and subsequently encoded using the quaternion encoder layer. The yielded multimodal feature in green facilitates visual and textual prompts in a seamless and efficient manner without requiring explicit interaction mechanisms.

## 3.2 Prompt Learning with Quaternion Networks

Our paper presents a ground-breaking technique called Prompt Learning with Quaternion Networks (QNet), which enables more effective encoding of multimodal features with fewer parameters. As shown in Figure 2, our method involves combining the output image features from a pre-trained model with the text context features. The combined features are then fed into the quaternion encoder, which generates a quaternion representation. The quaternion hidden space employs three mutually orthogonal imaginary axes to capture complementary information from various modalities, providing a more nuanced view of the interdependencies among different features. To ensure effective fusion of disparate modalities, our model employs multiple layers that facilitate the integration of hierarchical features inherent to different modalities. Finally, the fused features are used to create image and text prompts, which are subsequently fine-tuned for downstream tasks using QNet.

### 3.2.1 Quaternion Representation of Multimodal Features

A multimodal feature expressed by quaternion can be defined as $Q_f$, which is in a four-dimensional space known as the quaternion algebra $\mathbb{H}$:

$$Q_f = r1 + x\mathbf{i} + y\mathbf{j} + z\mathbf{k} \tag{1}$$

where $r, x, y$ and $z$ are real numbers, and $1, \mathbf{i}, \mathbf{j}$ and $\mathbf{k}$ are the quaternion unit basis. Our QNet uses quaternion-valued operations in each learning sub-process, unlike conventional fully-connected layers that use real-valued operations. The QNet transforms the input features into hypercomplex numbers. A quaternion dense layer operates on quaternions for all parameters, including inputs, outputs, weights, and biases. The quaternion algebra is maintained by manipulating real matrices (Gaudet & Maida, 2018).

When utilizing CLIP's multimodal pre-trained model in the zero-shot setting, the combination of visual and textual knowledge proves to be complementary. To determine the weights of various modal distributions, the $i$-axis and $j$-axis are employed, while the $k$-axis weight is maintained to balance the relationship between the two modalities. Each input vector of size $K$ and output vector of size $V$ is divided into two parts: $x_i$ for the first imaginary part and $y_j$ for the second imaginary part, resulting in a quaternion $Q = x_i + y_j$. It is worth noting that, despite not assigning a value to the $k$-axis, we still uphold its weight throughout the quaternion algebra. The $k$-axis acts as a balancing axis that dynamically balances the weights of the $i$-axis and $j$-axis. Meanwhile, in a real-valued space, a fully-connected layer performs a real-valued dot product between an input vector and a real-valued

$K \times V$ weight matrix. In QNet, this function is substituted with a quaternion-valued dot product that employs quaternion-valued matrices.

### 3.2.2 MULTIMODAL FUSION FEATURES

In this section, we will explain how we combine features from different modalities in the zero-shot learning setting. Drawing on insights from multimodal pre-trained models, including Oscar (Li et al., 2020) and ALBEF (Li et al., 2021), our analysis reveals a pronounced emphasis on visual features for the effective fusion of images and text. To get the most representative visual features, we use the final output of the vision branch of the multimodal pre-trained model. Co-CoOp also shows that these visual features can improve the generalization performance of the model. When it comes to the text features, we choose context tokens in zero-shot contexts. The rationale is that continuous vectors as prompts can encode text context information more efficiently with fewer parameters.

To ensure optimal exploitation of information from both models, we initially amalgamate the Meta-Image, $F_{image}$—derived post-dimension transformation via Meta-Proj—with the context tokens, $F_{text}$. This combination yields a rudimentary multimodal feature, denoted as $F_{multi}$:

$$F_{multi} = F_{image} + F_{text}. \tag{2}$$

Then, we send this multimodal feature $F_{multi}$ into the Quaternion Encoder layer $Q_{ua}(\cdot)$, projecting the input features to the quaternion hidden space.

$$Q_{multi} = Q_{ua}(F_{multi}). \tag{3}$$

Within the quaternion latent space, we assign distinct weights to the complex distribution between various modalities by selecting the $i$-axis and $j$-axis (among the four axes of $r, i, j$, and $k$, the $r$-axis predominantly serves as the energy axis, as real numbers multiplied by imaginary numbers still yield imaginary numbers). Simultaneously, we employ the imaginary $k$-axis as a balancing axis. This strategy effectively prevents scenarios where one modality distribution weight becomes excessively large, consequently overshadowing other modality information. For unseen classes, fine-grained features are particularly important, and the lack of these features leads to incorrect classification results. How to effectively fuse this fine-grained knowledge between different modalities becomes the key to improving CLIP in zero-shot scenarios. Based on this analysis, we initialize a quaternion-valued weight matrix $W$ utilizing the polar form of each weight $w$:

$$w = |w|e^{q_{multi}^{\triangleleft}} = \varphi q_{multi}^{\triangleleft},$$
$$\text{and } w_{\mathbf{i}} = \varphi q_{multi\mathbf{i}}^{\triangleleft}, \ w_{\mathbf{j}} = \varphi q_{multi\mathbf{j}}^{\triangleleft}, \ w_{\mathbf{k}} = \varphi q_{multi\mathbf{k}}^{\triangleleft}. \tag{4}$$

We use the well-known initialization criteria in (Glorot & Bengio, 2010) and (He et al., 2015) to obtain $\varphi$. $q_{multi}^{\triangleleft}$ is a normalized pure imaginary quaternion $q_{multi}^{\triangleleft} = 0 + x\mathbf{i} + y\mathbf{j} + z\mathbf{k}$, where $x\mathbf{i}, y\mathbf{j}$, and $z\mathbf{k}$ are uniformly sampled from $[0, 1]$ and then normalized. Detailed information about quaternion parameter initialization can be found in Appendix. Although the $k$-axis does not serve as the parameter axis in QNet, its weights contribute to the updating process of the weights associated with the $i$-axis and $j$-axis, as depicted in Figure 1(a). After passing through the quaternion hidden space, we obtain the final fused feature $Q_{multi}$. This feature preserves the multimodal information to the greatest extent and captures the complementary modality relationships through different imaginary axes of the quaternion:

$$Q_{fusion} = [w_i, w_j]Q_{mutil}, \tag{5}$$

where $[\cdot, \ \cdot]$ denotes the concatenation of $w_i$ and $w_j$.

### 3.2.3 VISION PROMPTING

Since multimodal feature $Q_{fusion}$ already contains rich image-text abstraction, we do not adopt any other complex structures to construct visual prompts. After performing a simple dimension change on $Q_{fusion}$, we directly concatenate it as a visual prompt to the visual transformer layer.

We define $M$ as a set of context vectors that can be learned. Then, we can obtain a set of learnable tokens, which have the same dimensionality as the word embeddings. For each layer $i$, we construct a prompt for the image branch: $m_i = \left\{ \acute{Q}_{fusion}^1, \acute{Q}_{fusion}^2, \cdots, \acute{Q}_{fusion}^M \right\}$. As a concatenation of the image context vectors and the multimodal features $Q_{fusion}$, the vision prompt for the $i$-th class of

Table 1: Comparison with existing methods in base-to-novel generalization on 11 datasets. The absolute gains compared to MaPLe are highlighted in red and blue.

| | Average | | | ImageNet | | | Caltech101 | | | OxfordPets | | |
|---|---|---|---|---|---|---|---|---|---|---|---|---|
| | Base | Novel | HM | Base | Novel | HM | Base | Novel | HM | Base | Novel | HM |
| CLIP (Radford et al., 2021) | 69.34 | 74.22 | 71.70 | 72.43 | 68.14 | 70.22 | 96.84 | 94.00 | 95.40 | 91.17 | 97.26 | 94.12 |
| CoOp (Zhou et al., 2022b) | 82.69 | 63.22 | 71.66 | 76.47 | 67.88 | 71.92 | 98.00 | 89.81 | 93.73 | 93.67 | 95.29 | 94.47 |
| Co-CoOp (Zhou et al., 2022a) | 80.47 | 71.69 | 75.83 | 75.98 | 70.43 | 73.10 | 97.96 | 93.81 | 95.84 | 95.20 | 97.69 | 96.43 |
| MaPLe (Khattak et al., 2022) | 82.28 | 75.14 | 78.55 | 76.66 | 70.54 | 73.47 | 97.74 | 94.36 | 96.02 | 95.43 | 97.69 | 96.43 |
| **QNet (Ours)** | **83.32** | **75.65** | **79.30** | **77.00** | **71.00** | **73.88** | **98.40** | **95.70** | **97.03** | **96.20** | **97.80** | **96.99** |
| | +1.04 | +0.51 | +0.75 | +0.34 | +0.46 | +0.41 | +0.66 | +1.34 | +1.01 | +0.77 | +0.04 | +0.41 |

| | StanfordCars | | | Flowers102 | | | Food101 | | | FGVCAircraft | | |
|---|---|---|---|---|---|---|---|---|---|---|---|---|
| | Base | Novel | HM | Base | Novel | HM | Base | Novel | HM | Base | Novel | HM |
| CLIP (Radford et al., 2021) | 63.37 | 74.89 | 68.65 | 72.08 | 77.80 | 74.83 | 90.10 | 91.22 | 90.66 | 27.19 | 36.29 | 31.09 |
| CoOp (Zhou et al., 2022b) | 78.12 | 60.40 | 68.13 | 97.60 | 59.67 | 74.06 | 88.33 | 82.26 | 85.19 | 40.44 | 22.30 | 28.75 |
| Co-CoOp (Zhou et al., 2022a) | 70.49 | 73.59 | 72.01 | 94.87 | 71.75 | 81.71 | 90.70 | 91.29 | 90.99 | 33.41 | 23.71 | 27.74 |
| MaPLe (Khattak et al., 2022) | 72.94 | 74.00 | 73.47 | 95.92 | 72.46 | 82.56 | 90.71 | **92.05** | 91.38 | 37.44 | 35.61 | 36.50 |
| **QNet (Ours)** | **74.20** | **74.90** | **74.55** | **97.00** | **75.20** | **84.72** | **91.10** | 92.00 | **91.55** | **38.80** | **35.90** | **37.29** |
| | +1.26 | +0.90 | +1.08 | +1.08 | +2.74 | +2.16 | +0.39 | -0.05 | +0.17 | +1.36 | +0.29 | +0.79 |

| | SUN397 | | | DTD | | | EuroSAT | | | UCF101 | | |
|---|---|---|---|---|---|---|---|---|---|---|---|---|
| | Base | Novel | HM | Base | Novel | HM | Base | Novel | HM | Base | Novel | HM |
| CLIP (Radford et al., 2021) | 69.36 | 75.35 | 72.23 | 53.24 | 59.90 | 56.37 | 56.48 | 64.05 | 60.03 | 70.53 | 77.50 | 73.85 |
| CoOp (Zhou et al., 2022b) | 80.60 | 65.89 | 72.51 | 79.44 | 41.18 | 54.24 | 92.19 | 54.74 | 68.69 | 84.69 | 56.05 | 67.46 |
| Co-CoOp (Zhou et al., 2022a) | 79.74 | 76.86 | 78.27 | 77.01 | 56.00 | 64.85 | 87.49 | 60.04 | 71.21 | 82.33 | 73.45 | 77.64 |
| MaPLe (Khattak et al., 2022) | 80.82 | **78.70** | 79.75 | 80.36 | 59.18 | 68.16 | 94.07 | **73.23** | 82.35 | 83.00 | **78.66** | 80.77 |
| **QNet (Ours)** | **81.10** | 78.60 | **79.83** | **80.70** | **59.50** | **68.50** | **96.60** | 73.10 | **83.22** | **85.40** | 78.50 | **81.80** |
| | +0.28 | -0.10 | +0.08 | +0.34 | +0.32 | +0.34 | +2.53 | -0.13 | +0.87 | +2.40 | -0.16 | +1.03 |

the image branch is now $z(x) = \left\{ \acute{Q}^1_{fusion}, \acute{Q}^2_{fusion}, \cdots, \acute{Q}^M_{fusion}, x \right\}$. In the vision branch of CLIP alongside the input image tokens, new learnable tokens are further introduced in deeper transformer layers of the image encoder $k(\cdot)$ up to depth $L$. Let $c_i$ be the word embedding(s) for the class name, $E_i$ be patch embeddings, and $[\cdot, \ \cdot]$ refer to the concatenation operation, we have:

$$
\begin{aligned}
[c_i, E_i, \ \_\_] &= k_i \left([c_{i-1}, E_{i-1}, \ \mathrm{m}_{i-1}]\right) & i = 1, 2, \cdots, L, \\
[c_j, E_j, m_j] &= k_j \left([c_{j-1}, E_{j-1}, \ \mathrm{m}_{j-1}]\right) & j = L+1, \cdots, N.
\end{aligned}
\tag{6}
$$

### 3.2.4 LANGUAGE PROMPTING

After obtaining high-quality multimodal features, we construct language context prompts with these features. Similarly, defining $g(\cdot)$ as the text encoder function, we construct a prompt for the text branch: $t_i = \left\{ \grave{Q}^1_{fusion}, \grave{Q}^2_{fusion}, \cdots, \grave{Q}^M_{fusion} \right\}$. The language prompt for the $i$-th class of text branch is embedded by $h_i(x) = \left\{ \grave{Q}^1_{fusion}, \grave{Q}^2_{fusion}, \cdots, \grave{Q}^M_{fusion}, c_i \right\}$. New learnable tokens are further introduced in each transformer block of the language encoder $g(\cdot)$ up to a specific depth $L$:

$$
[ \ \_\_, c_i] = g_i \left([t_{i-1}, c_{i-1}]\right) \ i = 1, 2, \cdots, L.
\tag{7}
$$

After the $L^{\text{th}}$ transformer layer, the subsequent layers process the previous layer prompts:

$$
[t_j, c_i] = g_j \left([t_{j-1}, c_{j-1}]\right) \ j = L+1, \cdots, N.
\tag{8}
$$

### 3.2.5 THE BRIDGE OF VISION LANGUAGE PROMPTING

The primary distinction of our approach from other multimodal prompts is that ours does not rely on explicit structural comparisons between different modalities. QNet capitalizes on a multimodal feature representation $Q_{multi}$ that encapsulates abundant information across various modalities. Consequently, it attains superior fusion performance compared to previous methods. Concretely, we utilize three mutually orthogonal imaginary axes to provide unique perspectives on multimodal features. We partition the input along the $i$-axis and $j$-axis to secure the image and text information,

Table 2: Comparison with existing methods in cross-dataset evaluation. We train QNet on the 1,000 classes of ImageNet and then transfer this model for evaluation on the remaining 10 datasets.

| | Source | Target | | | | | | | | | | |
|---|---|---|---|---|---|---|---|---|---|---|---|---|
| | ImageNet | Caltech101 | OxfordPets | StanfordCars | Flowers102 | Food101 | Aircraft | SUN397 | DTD | EuroSAT | UCF101 | Average |
| CoOp (Zhou et al., 2022b) | **71.51** | 93.70 | 89.14 | 64.51 | 68.71 | 85.30 | 18.47 | 64.15 | 41.92 | 46.39 | 66.55 | 63.88 |
| Co-CoOp (Zhou et al., 2022a) | 71.02 | **94.43** | 90.14 | 65.32 | 71.88 | 86.06 | 22.94 | 67.36 | 45.73 | 45.37 | 68.21 | 65.74 |
| MaPLe (Khattak et al., 2022) | 70.72 | 93.53 | 90.49 | 65.57 | 72.23 | **86.20** | **24.74** | 67.01 | 46.49 | 48.06 | 68.69 | 66.30 |
| QNet (Ours) | 71.10 | 94.30 | **90.80** | **66.00** | **73.10** | 86.10 | 24.20 | **67.90** | **46.90** | **51.80** | **68.70** | **66.98** |

respectively. The three imaginary axes possess equal status and select either the $i$-axis and $k$-axis or the $j$-axis and $k$-axis to produce equivalent results. Note that we do not assign a value to the $k$-axis relative to the $i$-axis and $j$-axis. Instead, we preserve the weight of the $k$-axis as a balance between the $i$-axis and $j$-axis. Our model effectively decouples the complex feature distributions of text and image by orthogonally disentangling them across the $i$-, $j$-, and $k$- axes. Moreover, by employing the quaternion network to integrate features across diverse hierarchical features of CLIP, QNet demonstrates its enhanced generalization capacity.

A substantial body of research concentrates on the development of explicit interaction structures between images and text. However, we argue that regardless of their complexity, these structures fail to surmount the limitations of homogeneous representation and effectively tackle the fundamental challenge in multimodal fusion: capturing intricate relationships among different modalities. Our approach employs quaternion networks which are adept at encoding interdependent relationships within multimodal features. This obviates the need to design extra interaction structures between images and text. Finally, the prediction probability is computed as:

$$p(y \mid \boldsymbol{x}) = \frac{\exp\left(\text{sim}\left(\boldsymbol{z}(Q(\boldsymbol{x})), g\left(\boldsymbol{h}_y(Q(\boldsymbol{x}))\right)\right)/\tau\right)}{\sum_{i=1}^{K} \exp\left(\text{sim}\left(\mathbf{z}(Q(\boldsymbol{x})), g\left(\boldsymbol{h}_i(Q(\boldsymbol{x}))\right)\right)/\tau\right)} \tag{9}$$

where $Q(\cdot)$ denotes the quaternion networks, $\mathbf{z}(x)$ is the vision prompt, $\boldsymbol{h}_i(x)$ is the language prompt, $\text{sim}(\cdot, \cdot)$ is the cosine similarity, and $\tau$ is a learned temperature parameter.

## 4 EXPERIMENTS

### 4.1 EXPERIMENTAL SETUP

**Datasets.** We follow Zhou et al. (2022b) by using 11 image recognition datasets that cover various tasks. Concretely, we include ImageNet (Deng et al., 2009) and Caltech101 (Fei-Fei et al., 2004) for generic object classification, Oxfordpets (Parkhi et al., 2012), StanfordCars (Krause et al., 2013), Flowers102 (Nilsback & Zisserman, 2008), Food101 (Bossard et al., 2014), and Aircraft (Maji et al., 2013) for fine-grained classification, SUN397 (Xiao et al., 2010) for scene recognition, UCF101 (Soomro et al., 2012) for action recognition, DTD (Cimpoi et al., 2014) for texture recognition, and EuroSAT (Helber et al., 2019) for satellite image recognition.

**Implementation details.** We evaluate our method in three scenarios: 1) Base-to-novel generalization, generalizing from base classes to new classes within a dataset; 2) Cross-dataset evaluation, transferring across different datasets, and 3) Domain generalization, transferring on four variant datasets of ImageNet. As for domain generalization, we conduct the experiments using ImageNet as the source dataset and four ImageNet variants that have different types of domain shift as the target datasets, namely ImageNetV2 (Recht et al., 2019), ImageNet-Sketch (Wang et al., 2019a), ImageNet-A (Hendrycks et al., 2021b), and ImageNet-R (Hendrycks et al., 2021a). For the training of QNet, we prompt-tune a pre-trained ViT-B/16 CLIP model and set prompt depth $L$ to 7 and language and vision prompt lengths to 2. We train QNet for 7 epochs with a batch size of 1 on a single NVIDIA RTX 8000 GPU. Following Zhou et al. (2022b), we use the pre-trained word embeddings of a shorter context (e.g., "a photo of ") to initialize the context vectors for the first layer. To maintain robust results, we validate our method using 16 shots and report the average results over three runs.

Table 3: Comparison with existing methods in domain generalization. We employ our method pre-trained on the source dataset (i.e., ImageNet) and test on four variant datasets of ImageNet.

| | Source | Target | | | | |
|---|---|---|---|---|---|---|
| | ImageNet | ImageNetV2 | ImageNet-S | ImageNet-A | ImageNet-R | Average |
| CLIP (Radford et al., 2021) | 66.73 | 60.83 | 46.15 | 47.77 | 73.96 | 57.18 |
| CoOp (Zhou et al., 2022b) | **71.51** | 64.20 | 47.99 | 49.71 | 75.21 | 59.28 |
| Co-CoOp (Zhou et al., 2022a) | 71.02 | 64.07 | 48.75 | 50.63 | 76.18 | 59.91 |
| MaPLe (Khattak et al., 2022) | 70.72 | 64.07 | 49.15 | 50.90 | 76.98 | 60.27 |
| QNet (Ours) | 71.10 | **64.30** | **49.20** | **51.30** | **77.70** | **60.65** |

## 4.2 BASE-TO-NOVEL GENERALIZATION

The baseline of our method is CLIP (Radford et al., 2021) with manual prompts. CoOp (Zhou et al., 2022b) improves CLIP with static prompts and CoCoOp (Zhou et al., 2022a) combines it with prompt-conditioned learning. MaPLe (Khattak et al., 2022) is a multimodal prompt method that achieves modality interaction through linear mapping, whereas our method constructs high-quality multimodal fusion features. Table 1 provides a comparative analysis of our QNet against other approaches in the base-to-novel generalization framework. Evidently, with fewer trainable parameters (i.e., 2.93M in QNet compared to 3.56M in MaPLe), QNet surpasses all previous prompt learning techniques in terms of base accuracy, novel accuracy, and harmonic mean (HM) when averaged across 11 datasets. Concretely, our method QNet attains a base category accuracy of 83.32%, elevates novel category performance from 75.14% to 75.65%, and increases the harmonic mean from 78.55% to 79.30%. The experimental outcomes effectively demonstrate that QNet associates semantic spaces across different modalities from multiple perspectives. For general datasets, QNet accomplishes comprehensive enhancements for both base and novel classes (i.e., 77.00% and 71.00% on ImageNet, 98.40% and 95.70% on Caltech). These general datasets encompass more diverse information than fine-grained datasets, necessitating the capture of different feature aspects via mutually orthogonal imaginary axes in the quaternion latent space. Linear mapping in MaPLe and other explicit interaction techniques are inadequate for addressing the shortcomings of homogeneous representation in capturing distinct aspects of data. To sum up, our method fundamentally addresses high-dimensional data processing, encoding the innate inter-dependencies within features in a more compact manner.

## 4.3 CROSS-DATASET EVALUATION

In the cross-dataset transfer setting, we first train QNet on the 1,000 classes of ImageNet and directly transfer the model to evaluate on the remaining 10 datasets. Table 2 presents comparative results of QNet and other methods. On the whole, QNet outperforms the baseline models substantially, attaining an average accuracy of 66.98%. Note that, during the training process on the 1,000 ImageNet classes, MaPLe employs a prompt depth of 3 while QNet integrates the quaternion networks solely within the initial two layers. Moreover, our experiments are conducted by preserving the same learning rate as in the base-to-novel generalization. The results suggest that our modality fusion approach outperforms MaPLe, achieving a more efficient encoding of rich multimodal information with fewer parameters and faster convergence.

## 4.4 DOMAIN GENERALIZATION

Evaluating models on out-of-distribution tasks is vital for assessing their generalization abilities. We employ our method pre-trained on the ImageNet dataset and subject them to direct testing on specific datasets with diverse data distributions. After completing the training on the entire 1,000 classes of ImageNet, we proceed to evaluate the model on four variant datasets derived from ImageNet, as depicted in Table 3. On the ImageNetV2 dataset, while both MaPLe and Co-CoOp yield an accuracy of 64.07%, our proposed method demonstrates superior performance with an accuracy rate of 64.30%. QNet consistently outperforms MaPLe and Co-CoOp across these four distinct ImageNet dataset variations. These results indicate that within the quaternion hidden space, QNet efficiently decouples the intricate distributions of distinct modalities by utilizing three mutually orthogonal imaginary axes, thereby correlating complementary information across modalities from diverse viewpoints.

Table 4: Ablation study among QNet, Co-CoOp, and MaPLe with one single layer.

|  | Base | Novel | HM |
|---|---|---|---|
| Co-CoOp | 80.47 | 71.69 | 75.83 |
| MaPLe (one layer) | 78.69 | 73.32 | 75.91 |
| QNet (one layer) | **81.01** | **74.88** | **77.83** |
|  | +2.32 | +1.56 | +1.92 |

Figure 3: Ablation study on the prompt depth of QNet on the Caltech dataset.

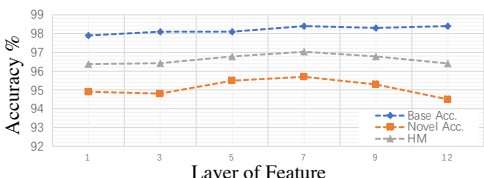

Table 5: Ablation study on adding real axis for modality fusion.

|  | Base | Novel | HM |
|---|---|---|---|
| $QNet_{r\,i}$ | 98.05 | 95.00 | 96.50 |
| $QNet_{r\,i\,j}$ | 98.10 | 94.33 | 96.18 |
| $QNet_{r\,i\,j\,k}$ | 97.93 | 93.90 | 95.87 |
| $QNet_{i\,j\,k}$ | **98.40** | **95.90** | **97.13** |

Table 6: Ablation study among different imaginary axes for modality fusion.

|  | Base | Novel | HM |
|---|---|---|---|
| $QNet_{i}$ | 98.23 | 94.93 | 96.55 |
| $QNet_{i\,j}$ | 98.10 | 94.17 | 96.09 |
| $QNet_{i\,j\,kp}$ | 98.07 | 95.17 | 96.60 |
| $QNet_{i\,j\,km}$ | **98.40** | **95.90** | **97.13** |

## 4.5 ABLATION STUDIES

**Multimodal fusion approach.** We further validate our innovative multimodal feature fusion approach by adjusting the prompt depth of both MaPLe and QNet to 1, while bearing in mind that the original Co-CoOp only has a single prompt layer. Our experimental findings, as indicated in Table 4, reveal that our method is more effective in modality fusion, with QNet showing a significant improvement compared to the other two techniques. MaPLe's lower base class accuracy compared to Co-CoOp indicates its improvements are largely driven by deep prompt strategy, rather than multimodal fusion. Our method adeptly captures the correlations among multimodal features, resulting in better performance. See the Appendix for complete results for each dataset.

**Effect of different axes.** We explore the impact of varying axes in our experiments and summarize the results in Tables 5 and 6. The study, using Caltech as an example, reveals a performance decline when the real axis is incorporated. We noted improved performance on novel classes with an increased number of imaginary axes; especially, using $k_m$ as a modulation axis proved more effective, enhancing the encapsulation of intermodal semantic spatial correlations between text and image modalities, compared to when $k_p$ serves as a parameter axis.

**Prompt depth.** Figure 3 demonstrates the effects of prompt depth on model performance. We again use the Caltech dataset to illustrate the influence of varying layers on the average performance. The overall trend suggests that performance improves with increasing model depth. However, it is observed that once the depth surpasses a certain threshold, performance begins to deteriorate, indicating a certain degree of overfitting caused by the increase in the number of layers. The optimal performance of our QNet is achieved with 7 layers.

## 5 CONCLUSION

Large-scale multimodal pre-trained models are a crucial step towards achieving general artificial intelligence. Both academic and industrial communities are focused on deploying these models for specific tasks. However, models like CLIP face challenges in handling fine-grained classifications and abstract tasks in zero-shot learning. To address this, we propose a fundamental high-dimensional structure using quaternion networks that capture rich semantic associations across different modalities and hierarchical features. In the quaternion latent space, the three mutually orthogonal imaginary axes are highly effective. Our experimental results demonstrate that QNet outperforms existing methods while involving fewer learnable parameters in base-to-novel generalization, cross-dataset transfer, and domain generalization. Furthermore, QNet effectively fuses features from multimodal sources in zero-shot scenarios, and can be extended to other domains.

**Limitation.** We use the CLIP model with transformer architecture for image feature extraction, but it tends to miss local and fine-grained details, impacting tasks such as classification and detection. Future plans include leveraging QNet to integrate these crucial features and addressing the transformer's bias towards semantic information.

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
