# PROMPT LEARNING WITH QUATERNION NETWORKS —SUPPLEMENTARY MATERIAL—

**Boya Shi[1,2], Zhengqin Xu[1], Shuai Jia[1], Chao Ma[1]***
[1] MoE Key Lab of Artificial Intelligence, AI Institute, Shanghai Jiao Tong University
[2] National Innovation Institute of Defense Technology
{boya.shi, fate311, jiashuai, chaoma}@sjtu.edu.cn

In this supplementary material, we provide more details and experimental results to enrich the manuscript. We first detail the initialization process of $\varphi$ based on the weight variance of QNet. Subsequently, we present the complete performances of QNet, MaPLe, and Co-CoOp on various datasets with a single-layer prompt setting. We then provide a performance comparison between QNet and cross attention, a representative method for explicit interaction, under optimal parameter settings. Furthermore, we validate the true source of the enhancement in model performance from various perspectives and provide a thorough analysis of model parameters and computational efficiency. Last but not least, we furnish a concrete demonstration of the performances of QNet and MaPLe through specific examples.

## A  PARAMETER INITIALIZATION OF QNET

In the QNet, the variance of $W$ is:

$$\text{Var}(W) = \mathbb{E}\left(|W|^2\right) - [\mathbb{E}(|W|)]^2, \text{ with } [\mathbb{E}(|W|)]^2 = 0. \tag{1}$$

Since the weight distribution is normalized, the value of $\text{Var}(W) = \mathbb{E}\left(|W|^2\right)$ is not trivial and $W$ follows a Chi-distribution with four degrees of freedom (DOFs), it can be further expressed as:

$$\text{Var}(W) = \mathbb{E}\left(|W|^2\right) = \int_0^\infty x^2 f(x)\mathrm{d}x = 4\sigma^2 \tag{2}$$

By following Glorot (Glorot & Bengio, 2010) and He (He et al., 2015), it can be extended to quaternion as: $\sigma = \frac{1}{\sqrt{2(n_{\text{in}} + n_{\text{out}})}}$, and $\sigma = \frac{1}{\sqrt{2n_{\text{in}}}}$, where $n_{\text{in}}$ and $n_{\text{out}}$ the number of neurons of the input and output layers respectively. Finally, we sample $\varphi$ from $[-\sigma, \sigma]$ to complete the weight initialization.

## B  MORE ABLATION EXPERIMENTS

**Multimodal fusion approach.** To further validate the effectiveness of our proposed novel approach in multimodal feature fusion, we have fixed the prompt depth at 1. We compare QNet with MaPLe and Co-CoOp within this specific setup. The experimental results on 11 diverse datasets are presented in Figure A(a), illustrating that even under a single-layer configuration, our method significantly outperforms Co-CoOp, with particularly notable improvements observed in the novel class. When applied to fine-grained datasets, such as EuroSAT and FGVC-Aircraft, our method realizes absolute performance gains that surpass 10%. Similarly, Figure A(b) reveals that our approach outperforms MaPLe with a single prompt layer across all 11 datasets. Notably, within the EuroSAT dataset, we record an absolute performance enhancement exceeding 9% for both base and novel classes. The experimental outcomes suggest that our approach efficiently disentangles the complex distributions of distinct modalities. This is achieved by utilizing three mutually orthogonal imaginary axes that correlate complementary information across varied modalities from diverse perspectives.

**Compared to cross attention method.** Cross attention, as a representative of explicit interaction structures, can dynamically adjust the weights of different modalities for the specific task, facilitating the adaptability of pre-training models to the current task. We also investigate the performance

---
*C. Ma is the corresponding author.

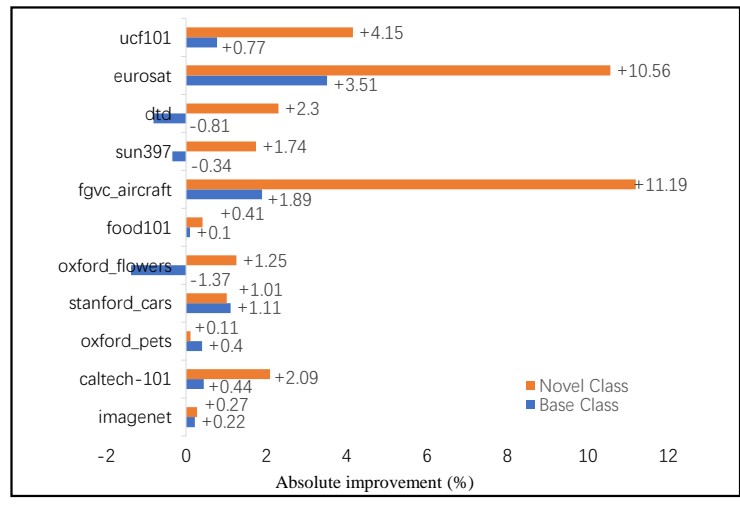

(a) QNet vs. Co-CoOp

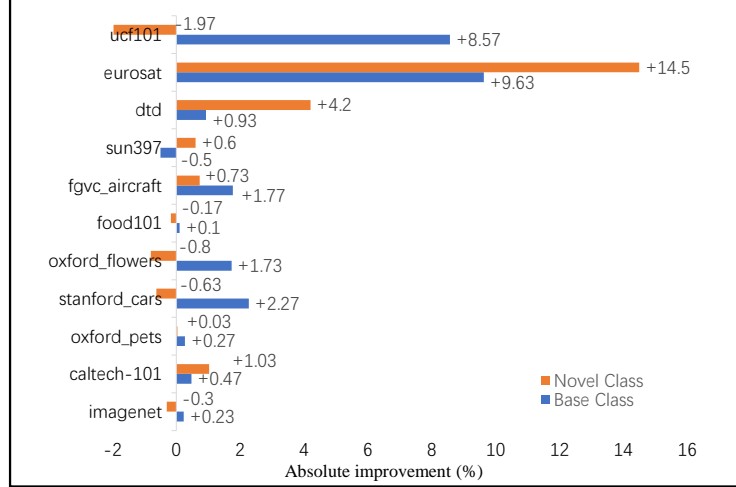

(b) QNet vs. Maple

Figure A: Ablation study among QNet, Co-CoOp, and MaPLe with one single layer.

of the cross attention method for modality fusion. After a search across various cross attention variants, Figure B displays a comparison of the performance between the cross attention method (i.e., with the optimal parameter configuration of two attention heads) and QNet across 11 datasets. Overall, QNet consistently outperforms cross attention. While their performance is on par in the ImageNet dataset, QNet continues to exceed the cross attention method in the novel class category on multiple fine-grained datasets. Most notably, in the novel class of the EuroSAT, QNet surpasses cross attention with an absolute gain of 17.2%. These experimental results suggest that the cross attention method fails to effectively capture distinct aspects and patterns in the data. This strategy involves assigning maximum weight to the modality that is beneficial to the current task. However, it fails to fully leverage the rich complementary information among different modalities, thereby inadvertently discarding essential data. In contrast, QNet demonstrates superior efficacy in multimodal fusion, notably enhancing the efficiency of multimodal prompt learning.

In addition, we have also explored various fusion methods under our architecture, including concatenating features, MLP and adapter. Specifically, the adapter method involves trainable feed-forward neural networks to first reduce the dimensionality of the input vector and then restore its dimensionality. Taking the Caltech dataset as an example, the results for various fusion methods are illustrated in Table A. We find that the proposed QNet achieves greater performance than other fusion methods,

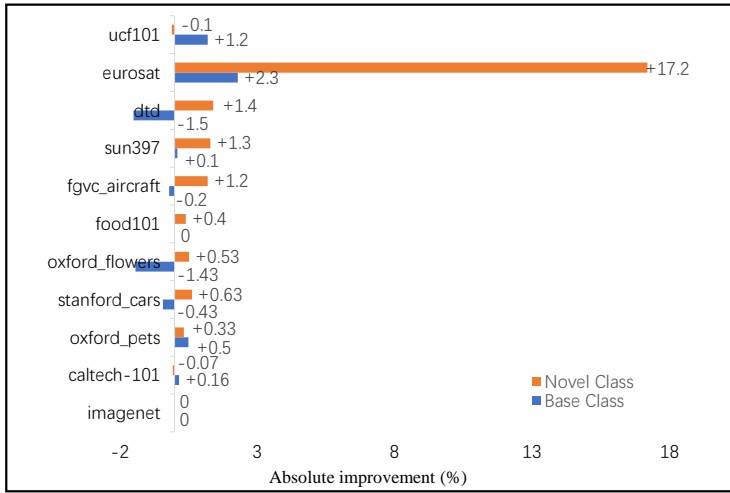

Figure B: Ablation study on investigating cross attention method and QNet.

especially for the novel classes. Experiments show that our method can better capture semantically interrelated features between different modalities. We will supply the complete ablation study on various modality fusions.

Table A: Ablation study among different methods for modality fusion.

| Modality fusion | Base | Novel | HM |
|---|---|---|---|
| Concatenation | 98.00 | 94.03 | 95.97 |
| MLP | 97.90 | 94.17 | 96.00 |
| Adapter | 97.95 | 94.43 | 96.16 |
| QNet | **98.40** | **95.90** | **97.13** |

**Parameters and FLOPs.** For fair comparisons, we first report the parameters and GFLOPs of MaPLe and our method, as they both adopt multimodal prompts (i.e., text and images) with multiple layers. The total parameters in MaPLe and ours are 127.88M and 127.25M, while the trainable parameters in MaPLe and ours are 3.56M and 2.93M, respectively. It is observed that our method uses fewer parameters but achieves better performance across 11 datasets in the base to novel scenarios, cross-dataset evaluation, and domain generalization scenarios. On the other hand, MaPLe has 167 GFLOPs while ours is 179 GFLOPS, indicating our method is better than MaPLe in terms of parameter efficiency. Additionally, the total parameters of our method (i.e., 127.25M) are slightly larger than CoOp (i.e.,125.14M) and Co-CoOp (i.e.,124.36M), as both CoOp and Co-CoOp only use text prompts with a single layer, leading to weak performance.

**Source of improvement.** A noteworthy observation was made: when the real axis of the two modalities was nullified, there was a general uptick in performance. It is postulated that this enhancement in performance could be attributed to the reduction of redundant information from one modality, thus balancing the data weight across different modalities.

To explore this further, we applied QNet to the two streamlined modality representations. As outlined in Table B, taking the Caltech dataset as an example, the streamlined features undeniably enhance performance. Simultaneously, the QNet fusion between modalities offers additional benefits. This ablation study stands as a testament to our approach's proficiency in bridging the gap between different modalities. Further, we individually mapped each modality through QNet and subsequently concatenated the final vectors. By applying the QNet to these two modality representations independently to investigate if it can focus more on only extracting these modalities' interactions. It becomes evident through our experiments that our method is adept at discerning the complex interrelations between different modalities within the Quaternion space rather than reducing redundant information from one modality.

Table B: Ablation study among different methods for modality fusion.

| Method | Base | Novel | HM |
|---|---|---|---|
| $QNet_{clean}$ | 98.07 | 94.73 | 96.37 |
| $QNet_{individual}$ | 98.30 | 95.90 | 97.09 |
| QNet | **98.40** | **95.90** | **97.13** |

**Expanded comparisons and evaluations.** Further, we extend our QNet to related tasks, such as video understanding. We conduct preliminary experiments on the UCF-101 video dataset (see Table C). Using the ViFi-CLIP's base-to-novel generalization setting, QNet was applied to a Kinetics-400 pre-trained ViFi-CLIP with prompt learning. QNet outperforms the IVLP method and even exceeds fully fine-tuned models like ActionCLIP, indicating its strong generalization ability across various modalities, including video tasks.

| Method | Base Acc. | Novel Acc. | HM |
|---|---|---|---|
| Vanilla CLIP | 78.50 | 63.60 | 70.30 |
| ActionCLIP | 85.60 | 75.30 | 80.10 |
| XCLIP | 95.40 | 74.00 | 83.40 |
| A5 | 95.80 | 71.00 | 81.60 |
| IVLP | 95.90 | 74.10 | 83.60 |
| QNet | **96.60** | **78.20** | **86.43** |

Table C: Performance comparison in video action recognition generalization benchmark on UCF-101. We employ QNet and IVLP on ViFi-CLIP and compare with the prior video approaches.

## C  ENHANCED GENERALIZATION CAPABILITY

In this section, we primarily illustrate the modality fusion capability of QNet by showcasing specific examples. As shown in Figure C(a), we can observe from the examples that even in the base classes, MaPLe struggles to handle challenges such as intra-class variation (e.g., cups of the same category stacked together), object occlusion (e.g., an electric guitar with only the neck visible and an airplane with only half the fuselage visible), and background clutter (e.g., a hawksbill with people and a camera in the background). These examples highlight the limitations of the MaPLe when confronted with the complex variations and distractions present in real-world scenarios. In contrast, QNet, with its richer multimodal fusion features, can effectively address the challenges of intra-class variation, object occlusion, and background clutter in recognition tasks.

In novel classes, as shown in Figure C(b), MaPLe still performs poorly in recognition scenarios involving intra-class variation (e.g., viewpoint changes in kangaroos) and image quality (e.g., noise-included scissors). Notably, even for classes with good image quality and front-facing views (e.g., Snoopy and Ketch), MaPLe still misclassifies. This further illustrates the importance of the QNet's fusion features in enhancing generalization capabilities.

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

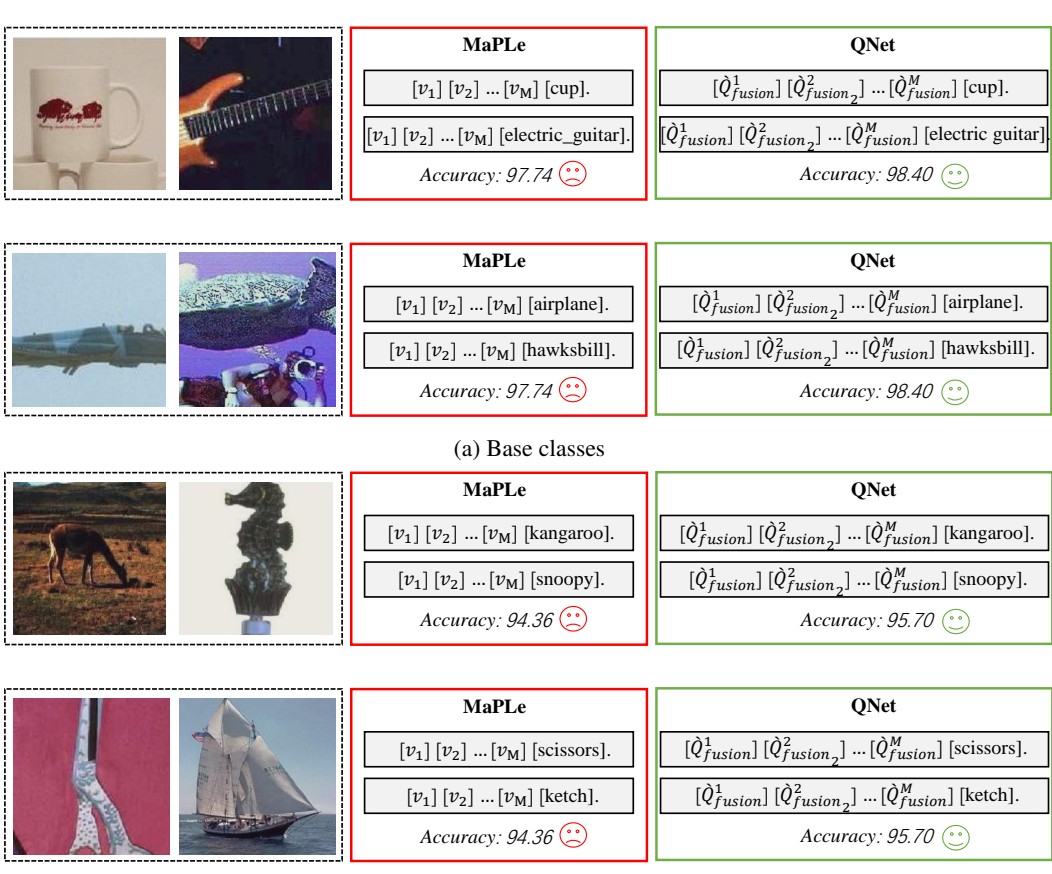

Figure C: Examples of Base-to-Novel Generalization. The images were randomly selected from the general dataset, Caltech101. These instances showcase the ability of QNet to recognize and encode complex multi-modal structures and contexts.