# OpenReview forum: "Prompt Learning with Quaternion Networks"
_ICLR.cc/2024/Conference — ICLR 2024 poster_

### Official Review · Reviewer_Rjda · 2023-10-29

**Soundness:** 4 excellent
**Presentation:** 3 good
**Contribution:** 4 excellent
**Rating:** 8
**Confidence:** 3

**Summary:**

This paper devised a method that combines a pre-trained model to achieve high performance in VL tasks even in situations where there is no training data. It also achieved excellent benchmark results in three validation tests: base-to-novel generalization, cross-dataset transfer, and domain transfer scenarios, even compared to MaPLE, one of the latest models.

**Strengths:**

Benchmark tests have been conducted and it has achieved excellent results compared to SOT methods. The benchmarks are also reasonable and provide excellent comparisons.

**Weaknesses:**

The structure of the model shown in Figure 2 is poorly explained, making it difficult to understand the difference between it and other models. I also got the impression that there was a lack of consideration as to why MaPLE achieved such excellent results. I would have liked a more detailed chapter to explain why this model is so good.

**Questions:**

None

---

> ### Author Response · Authors · 2023-11-22
> **Response to Reviewer Rjda**
>
> We appreciate your valuable feedback and address your concerns as follows.
>
> **W: Model structure and performance.**
> As suggested, we will update Figure 2 to supply more detailed comparisons to help readers understand the differences between our method and other approaches. In general, both MaPLE and our proposed QNet are based on the architecture of CLIP. MaPLE primarily employs linear projections to align and merge features from different modalities. Linear projection can maintain mutual synergy and reduce the tendency to learn independent uni-modal prompts, proving effective in specific situations. However, it oversimplifies the intricate interconnections in multimodal data. In our approach, QNet handles diverse modalities by leveraging the inherent orthogonal relationship among the three imaginary axes within the quaternion hidden space. Unlike linear projection in MaPLE, our QNet utilizes the quaternion space to construct the intricate and non-linear associations between modalities, which yields better performance in multiple datasets, especially in zero-shot learning scenarios.

---

### Official Review · Reviewer_jgw2 · 2023-10-30

**Soundness:** 3 good
**Presentation:** 3 good
**Contribution:** 3 good
**Rating:** 6
**Confidence:** 4

**Summary:**

This work aims to improve the performance of a multi-modal pre-trained foundation model via prompt tunning. This work proposes to use Quaternion Networks to align the semantics across modalities while finetuning. Quaternion Network projects feature quaternion hidden space, where three mutually orthogonal imaginary axes, namely i, j, and k, allocate unique weights to various distribution features from diverse perspectives. Compared to previous prompt learning works, the major difference is introducing quaternion hidden space to fuse data modalities. This work conducts experiments on more than 10 datasets which is solid to some extent.

Pros:
- This work introduces quaternion hidden space to prompt learning for foundation models, which is new.
- Experiments cover a wide range of datasets.

Cons:
- Comparing the proposed method with previous prompt learning methods on computation overhead and latency is needed.
- Quaternion hidden space seems to be more sophisticated than linear space which might be better than linear projection. However, it's not obvious why Quaternion Networks is better than the previous prompting technique; or why tunning multimodal pre-trained networks needs quaternion hidden space.


In-depth comparison with previous prompting methods and analysis of this quaternion network improve this work.

**Strengths:**

Pros:
- This work introduces quaternion hidden space to prompt learning for foundation models, which is new.
- Experiments cover a wide range of datasets.

**Weaknesses:**

Cons:
- Comparing the proposed method with previous prompt learning methods on computation overhead and latency is needed.
- Quaternion hidden space seems to be more sophisticated than linear space which might be better than linear projection. However, it's not obvious why Quaternion Networks is better than the previous prompting technique; or why tunning multimodal pre-trained networks needs quaternion hidden space.

**Questions:**

-

---

> ### Author Response · Authors · 2023-11-22
> **Response to Reviewer jgw2**
>
> We appreciate your valuable comments and address your concerns as follows.
>
> **Cons1: Computation overhead and latency.**
> For a fair comparison, we calculated the parameters and GFLOPs for both MaPLe and our proposed QNet, which use multimodal prompts. Our method involves fewer trainable parameters (2.93M) than MaPLe (3.56M) yet achieves better performance across 11 datasets. Regarding computational overhead, while MaPLe operates at 167 GFLOPs, our method runs a slightly higher computational load at 179 GFLOPs. This slight increase is attributed to the more complex non-linear quaternion space compared to the linear space of MaPLe. As for latency, CoOP, Co-CoOp, Maple, and our QNet runs at 299Fps, 48Fps, 103Fps, and 50Fps under processing a hundred images per batch. Among these methods, MaPLe and CoOp exhibit higher speeds as they only require a single forward pass of prompts through the text encoder. In contrast, Co-CoOp and our QNet, which rely on instance-conditional designs, demand an independent forward pass for instance-specific prompts. Hence, Co-CoOp and our QNet consume more GPU memory when the batch size is larger than 1, leading to slightly slower running speeds.
>
> **Cons2: The superiority of quaternion spaces.**
> We acknowledge that quaternion hidden spaces are more sophisticated than linear spaces, resulting in better performance. However, current multimodal fusion methods often rely on explicit interaction structures that struggle to fully capture the varied and complex patterns inherent in multimodal data. For example, MaPLE with linear mappings oversimplifies the associations between different modalities. Similarly, the cross-attention method assigns maximum weight to the modality that is beneficial to the current task, but fails to leverage the rich complementary information among different modalities, inadvertently discarding essential data. We have presented detailed ablation experiments in Appendix Figure B. In our approach, QNet, we use quaternion hidden space to construct more complex and non-linear relationships among diverse modalities in a more compact manner. First, we merge output image features from a pre-trained model with text context features. Then, we process these integrated features through a quaternion encoder. Within the quaternion hidden spaces, our QNet employs orthogonal imaginary axes (i.e., i, j, and k) to further tackle the fused features in a higher dimension. Our approach provides a holistic understanding and captures intricate details that might be overlooked from a single-dimensional perspective. Additionally, our QNet integrates features across hierarchical levels to enrich the representation from various perspectives. Extensive experiments on multiple datasets demonstrate the superiority of our approach over the state-of-the-art methods. Furthermore, our QNet can be used as a plug-and-play module to support various multimodal approaches.

---

> > ### Comment · Reviewer_jgw2 · 2023-12-05
> > **Response**
> >
> > My concerns are solved. And I will raise my score.

---

### Official Review · Reviewer_xbTq · 2023-11-19

**Soundness:** 3 good
**Presentation:** 3 good
**Contribution:** 3 good
**Rating:** 6
**Confidence:** 4

**Summary:**

The paper discusses the challenges and limitations of multimodal pre-trained models in capturing diverse and complementary features across different modalities. It introduces a novel approach called QNet, which utilizes quaternion networks to improve the modality fusion capacities of pre-trained models.

**Strengths:**

The use of quaternion networks to capture the intricate relationships among different modalities is a novel idea that sets this paper apart from existing methods. The paper provides a thorough analysis of the proposed method, including experimental results on various datasets and comparison with existing methods. The results are presented in a clear and concise manner.

The paper is well-written and organized, making it easy to follow the proposed approach and understand the experimental results.

**Weaknesses:**

Overall, this paper presents a sound framework. My main concern is that the authors should compare to the baseline scombining Quaternion Networks and the existing prompt learning method clearly. Besides, the benefits of QNet can be evaluated on more multimodal tasks.

**Questions:**

Please see my comments on the weaknesses.

---

> ### Author Response · Authors · 2023-11-23
> **Response to Reviewer xbTq**
>
> We appreciate your valuable feedback and address your concerns as follows.
>
> **W: Expanded comparisons and evaluations.**
> Our method, QNet, significantly differs from directly combining Quaternion networks and the existing prompt learning approach, where all real and imagery axes are used as parameter axes. Instead, our QNet only uses imagery axes in which the i-axis and j-axis act as parameter axes, and the k-axis acts as a modulation axis to balance the weights of the i-axis and j-axis. Experiments in Table 5 and Table 6 demonstrate the effects of different real and imagery axes, affirming that using imagery axes can better learn multimodal information from the fused features. We also compared our method with Coop and MaPle, both with the setting of a single layer in Table 4, which further validates the effectiveness of modality fusion in QNet. As suggested, we extend our QNet to related tasks, such as video understanding. Due to limited time, we conducted preliminary experiments on the UCF-101 video dataset (see Table 1). Using the ViFi-CLIP's base-to-novel generalization setting, QNet was applied to a Kinetics-400 pre-trained ViFi-CLIP with prompt learning. QNet outperforms the IVLP method and even exceeds fully fine-tuned models like ActionCLIP, indicating its strong generalization ability across various modalities, including video tasks. Moreover, we will make more detailed comparisons with other methods and conduct more evaluations on other tasks to validate the effectiveness of QNet in the revised version. We gratefully appreciate your suggestions to improve this work.
> | Method       | Base Acc. | Novel Acc. | HM    |
> |--------------|-----------|------------|-------|
> | Vanilla CLIP | 78.50     | 63.60      | 70.30 |
> | ActionCLIP   | 85.60     | 75.30      | 80.10 |
> | XCLIP        | 95.40     | 74.00      | 83.40 |
> | A5           | 95.80     | 71.00      | 81.60 |
> | IVLP         | 95.90     | 74.10      | 83.60 |
> | **QNet**     | **96.60** | **78.20**  | **86.43** |
>
> _Table 1: Performance comparison in video action recognition generalization benchmark on UCF-101. We employ QNet and IVLP on ViFi-CLIP and compare with the prior video approaches._

---

### Meta-Review · Area_Chair_GeNF · 2023-12-07

**Metareview:**

The paper proposes a new method based on Quaternion Networks for aligning semantics across multiple modalities during finetuning. The main goal is to better the performance of a multimodal foundation models with prompt tuning. All the reviewers are in agreement that the work is novel, and the experiments indicate the effectiveness of the approach. There is some criticism about clarity and exposition of the method.

I recommend accepting the paper based on the reviewer comments and suggest the authors to consider reviewer comments to make the exposition clearer.

**Justification For Why Not Higher Score:**

I recommend accepting as a poster because of reviewer criticism about experimentation and clarity.

**Justification For Why Not Lower Score:**

Reviewers are in agreement about the novel contribution.

---

### Decision · Program_Chairs · 2024-01-16

Accept (poster)